# Intelligent Predictive Solution Dynamics for Dahl Hysteresis Model of Piezoelectric Actuator

**DOI:** 10.3390/mi13122205

**Published:** 2022-12-12

**Authors:** Sidra Naz, Muhammad Asif Zahoor Raja, Ammara Mehmood, Aneela Zameer Jaafery

**Affiliations:** 1Department of Electrical Engineering, Pakistan Institute of Engineering and Applied Sciences, Islamabad 45650, Pakistan; 2Future Technology Research Center, National Yunlin University of Science and Technology, Yunlin 64002, Taiwan; 3School of Engineering, RMIT University, Melbourne 3001, Australia; 4Department of Computer and Information Sciences, Pakistan Institute of Engineering and Applied Sciences, Islamabad 45650, Pakistan

**Keywords:** piezoelectric actuator, Levenberg–Marquardt, Bayesian Regularization, intelligent computing, dahl hysteresis model

## Abstract

Piezoelectric actuated models are promising high-performance precision positioning devices used for broad applications in the field of precision machines and nano/micro manufacturing. Piezoelectric actuators involve a nonlinear complex hysteresis that may cause degradation in performance. These hysteresis effects of piezoelectric actuators are mathematically represented as a second-order system using the Dahl hysteresis model. In this paper, artificial intelligence-based neurocomputing feedforward and backpropagation networks of the Levenberg–Marquardt method (LMM-NNs) and Bayesian Regularization method (BRM-NNs) are exploited to examine the numerical behavior of the Dahl hysteresis model representing a piezoelectric actuator, and the Adams numerical scheme is used to create datasets for various cases. The generated datasets were used as input target values to the neural network to obtain approximated solutions and optimize the values by using backpropagation neural networks of LMM-NNs and BRM-NNs. The performance analysis of LMM-NNs and BRM-NNs of the Dahl hysteresis model of the piezoelectric actuator is validated through convergence curves and accuracy measures via mean squared error and regression analysis.

## 1. Introduction

Piezoelectric actuators (PEAs) are extensively utilized in precision positioning systems owing to their minor sizes, low noise and heat, high displacement resolution, high positioning accuracy, high energy density, rapid frequency response, and large force generation [1,2,3,4]. All of these benefits make PEAs universally exploited in a variety of fields, including vibration monitoring [5], machining [6], micro/nano observation and operation [7,8], calibration of optical fibers [9], hydraulic pipeline systems [10], and laser focusing mechanisms [11]. Precision positioning systems employ PEAs as they have properties such as free lubrication, free friction, and high resolution, and thus are often paired with consenting mechanisms [12]. However, a high range of inputs and low frequencies used for PEA actuation yield nonlinear hysteresis. As a result, these nonlinear hysteresis effects significantly reduce positioning precision. Therefore, the study of nonlinear hysteresis behaviors of the input voltage with output displacement is worthwhile and important to increase the nano/micro-positioning system’s accuracy. These hysteresis behaviors may be categorized into rate-independent and rate-dependent. In rate-independent hysteresis, both input voltage and output displacement are not linear with each other at a low frequency, while in rate-dependent hysteresis, the input frequency affects the hysteresis curves superficially, varying with the increase in frequency [13].

The initial necessary step to deal with hysteresis is to model it accurately and then compensate for this nonlinearity by using control mechanisms. For this purpose, different hysteresis models for PEAs were presented over the past few decades, i.e., the Prandtl Ishlinskii model, Krasonsel’skii Pokrovskii model, Preisach model, Polynomial-based model, Maxwell model, and Jiles Atherton model. These models were used to study the rate-independent hysteresis behaviors of PEAs. However, for rate-dependent hysteresis behavior studies, the Bouc Wen model, Duhem model, Dahl model, and Backlashnlike model were employed. All these models are further classified into two categories, physical and mathematical hysteresis models. Physical-based models involve the theoretical study of modeling and controlling of PEAs and exhibit quite complex structures owing to their intrinsic mechanism of involving finite element modeling which causes high computational cost, as in the Jiles Atherton model [14]. Mathematical models include the polynomial approximation [15], Preisach model [16], Duhem model [17], Prandtle Ishlinskii model [18], Maxwell slip model [19], LuGre model [20], Dahl model [21], and Bouc Wen model [22,23].

### 1.1. Related Study

In recent years, researchers presented numerous methodological strategies to study the parameters of the nonlinear hysteresis model resulting as a consequence of input voltage and output displacement to achieve maximum positioning accuracy. The feedforward control approach for output force is presented, which is based on the dynamic hysteresis inverse model to efficiently reduce the PEA’s nonlinear properties [24]. A sliding mode controller for the feedforward error reduction of the modified Prandtl Ishlinskii hysteresis model was presented [25]. A dynamic compensator was proposed to suppress the nonlinearity of nano-positioning [26]. Backstepping control integral sliding mode methodology was presented for precision motions [27]. A quasi-Rayleigh model was used to study the hysteresis of PEAs [28]. A system-level approach was proposed to study hysteresis behavior at any frequency [29]. A single neuron controller based on Hebb learning rules was used for error adjusting of unsymmetrical behavior [30]. A system depending on the frequency of the nonlinear hysterical effect based on the gated recurrent unit and neural Turing machine was presented [31,32]. To achieve high precision tracking that deals with system uncertainties, a self-tuning control-based neural network technique was presented [33]. A proportion integral differential control system was proposed to achieve higher accuracy of the static model [34]. Some parameter identification techniques such as the Transitional Markov Chain Monte Carlo approach [35] and the least squares-based method [36] were used for PEA hysteresis models. Furthermore, in the literature, the characteristics of some nature-inspired algorithms have also been exploited for the optimization of model parameters of PEAs. They include particle swarm optimization [37], clonal selection mechanism [38], genetic algorithms [39], artificial bee colony algorithm [40], and hybrid differential evolution and Jaya algorithm [41].

Differential system problems are frequently solved using numerical methods based on the soft computing paradigm [42,43]. The most recent and important studies include the solution of a mathematical model for nonlinear oscillatory Vander Pol Mathieu’s systems [44,45], Painlevé equation-II for asymmetric optical prototypes [46], a transport model designed for fluid and soft tissues solute as well as microvessels [47], the Lane Emden equation for astrophysics [48], nonlinear models of circuit theory [49], fuel ignition model [50], nonlinear Bratu equation-based models in electrical conductors [51,52], nanofluidic problems comprising nanotubes of carbon [53], financial paradigms [54], dusty*plasma [55], atomic physics [56], drainage problems [57], wind power [58], heartbeat dynamics [59], search space reduction of economic load dispatch problem [60], HIV-infected cells model [61], piezoelectric devices [62,63,64], fluidic flow models [65,66], fractional dynamic modeling equations [67], bilinear systems [68], multi-frequency response signals [69], entropy generation [70,71], Ree Eyring nanofluid flow [72], fractional order systems [73], state space systems [74], magnetohydrodynamics [75], stochastic systems [76], and data filtering [77].

### 1.2. System Model: Dahl Hysteresis Modeling of the Piezoelectric Actuator

Among all hysteresis models, the Dahl hysteresis model is the most simplified and closer depiction of the hysteresis loop with a lower number of parameters and better capturing of non-symmetric behavior of input voltage and output displacement [78]. Although well known for modeling friction, the Dahl model’s application towards piezoelectric hysteresis modeling seems to have a lot to give. A modified Dahl model for hysteresis providing estimates on observer design is presented in [79,80].

In this research study, the nonlinearity of the system is represented using a second-order Dahl model with fewer parameters, as shown in Equation (1).
(1)$$m\ddot{d}(t)+\varepsilon \dot{d}(t)+gd(t)=kv(t)-F_{h},$$
where *m* signifies the system mass, ε is the damping coefficient, *g* indicates the stiffness coefficient, *k* is the piezoelectric coefficient, *v*(*t*) denotes input voltage, *d* symbolizes the output displacement, and *F_h_* represents the nonlinear hysteresis force. The nonlinear force that involves the hysteresis parameters is mathematically represented in Equation (2).
(2)$$F_{h}=a_{1}b_{1}+b_{0}a_{2}\mathrm{sgn}(\dot{d}(t)),$$

Combining Equations (1) and (2) gives Equation (3):(3)$$\left(m\ddot{d}(t)+\varepsilon \dot{d}(t)+gd(t)-kv(t)+a_{1}b_{1}-b_{0}a_{2}\mathrm{sgn}(\dot{d}(t))\right)=0$$
with initial conditions $d(t)=0,\;\mathrm{and}\;\dot{d}(t)=0.$

The values of each of these dynamic and hysteresis model parameters are presented in Table 1. Figure 1 shows the graphical model of PEAs.

### 1.3. Problem Statement and Significance

The growing interest in PEAs has led to the implementation of a variety of approaches to observe dynamic hysteresis behavior. Soft computing approaches have not yet been thoroughly investigated for the interpretation of the Dahl hysteresis model. The major focus of this research was to employ soft computing paradigms for finding the solution of a Dahl hysteresis model of PEAs represented in Equation (3) using feedforward and backpropagated neural networks based on Levenberg–Marquardt and Bayesian Regularization training algorithms.

### 1.4. Contribution and Innovative Insights

The following are the innovative features of the proposed computing mechanism:The Dahl hysteresis model for PEAs is studied using a novel neuro intelligent and heuristic methodology based on training algorithms: Levenberg–Marquardt and Bayesian Regularization backpropagated neural networks.For training, testing, and validation of the presented model, the dataset of the Dahl hysteresis model is created by exploiting the efficiency of the Adams numerical approach.The ability of the presented methodology structure is corroborated on accuracy by the disparity in applied input voltage signals to the piezoelectric actuator Dahl model employing performance analyses based on regression analyses and mean squared error.In addition to the established skill of accurate solution, supplementary appreciated key properties of the presented scheme include extensive methods, easy implementation, speedy and stable convergence, constancy, and adaptability.

### 1.5. Organization

The remainder of this paper is set out as follows: Section 2 provides a proposed methodology along with the Adams numerical method, Levenberg–Marquardt backpropagation method, Bayesian Regularization method, and performance indices. Section 3 provides the results and discussion of all experiments carried out for the Dahl hysteresis system of a dynamic piezoelectric actuator. The conclusion of the proposed work is provided in Section 4.

## 2. Methodology

In this segment, the proposed methodology is described in three steps. First, the sample dataset of the Dahl hysteresis model is created by the Adams numerical method, and then model approximation is carried out by using an artificial neural network where optimization is performed by backpropagated networks using Levenberg–Marquardt method neural networks (LMM-NNs) and Bayesian Regularization method neural networks (BRM-NNs). Finally, the performance matrices are used to analyze the present study. Figure 2 represents the graphical abstract of the design methodology.

### 2.1. Adams Numerical Method

In this study, the dataset for the Dahl hysteresis model is created via the statistical solver, the Adams numerical scheme, to determine the approximate outcomes of PEAs signified in Equation (3) and all various scenarios presented in numerical Equations (11), (13), (15), and (17). The Adams numerical method is executed by using the “NDSolve” routine in the Mathematica platform to find the numerical solution of differential Equations (4)–(7) for obtaining the numerical solution of the Dahl hysteresis model of PEAs. The Adams method is a numerical approach to solving linear first-order systems of Equations (4) and (5).
(4)$$\frac{dy}{dx}=f(x,y)$$
(5)$$Y_{l+1}=y_{l}+\int_{t_{l}}^{t_{l+1}}\frac{dy}{dx}dt\;=\;y_{l}+\int_{t_{l}}^{t_{l+1}}f(y,t)\;dt$$
where *y* indicates the result of an ordinary differential equation (ODE), *x* signifies input data, $Y_{l+1}$ represents first-order interpolated iterative technique, and the term *t* represents the time interval of Adams approaches. These mathematical notations are constructed on the basic supposition of approximating the integral inside the period (*t_l_*, *t_l_*_+1_) using a polynomial.

Adams numerical methods are categorized into two types: the Adams Bash forth (AB), known as the explicit type, and the Adams Moulton (AM), known as the implicit type. The first-order AB and AM methods are the forward and backward procedures. The second-order Adams Bash forth (AB2) method and the Adams Moulton (AM2) method obtained by linear interpolation are often used and are described in Equations (6) and (7), where *q* represents the step interval.
(6)$$Y_{l+1}=y_{l}\;+\;\frac{q}{2}(3f(y_{l},t_{l}\;-\;f(y_{l-1},t_{l-1})),$$
(7)$$Y_{l+1}=y_{l}\;+\;\frac{q}{2}(f(y_{l+1},t_{l+1}\;+\;f(y_{l},t_{l})).$$

Artificial neural networks are made up of artificial neurons which are joined together. A piece of information is transferred to the next neuron through the connection between them. A neuron’s status is commonly stated as a series of real numbers ranging from 0 to 1. When learning progresses, the weight of neurons and synapses may alter, and as a result, an increase or decrease may occur in the strength of the signal which is transmitted downstream.

Normally, neurons are organized in layers where each layer can execute different kinds of transformation based on its input. This strategy deployed 100 hidden layers to achieve the desired result. Figure 3 depicts the fundamental architecture of the proposed system neural network. This architecture consists of two layers the hidden layer and output layer, where a sigmoid function is used as an activation function, and the two training algorithms, Levenberg–Marquardt and Bayesian Regularization, are used as backpropagated neural networks.

### 2.2. Levenberg–Marquardt Backpropagation Method

In this research, the Levenberg–Marquardt backpropagation neural network is used to find an optimized solution for the Dahl hysteresis model of the piezoelectric actuator. The dataset gathered by the Adams numerical method was used as a target output for LMM-NNs. In this methodology, the LMM-NNs algorithm was implemented by “nftool” in MATLAB for machine learning. This algorithm was developed by Kenneth Levenberg and Donald Marquardt for the first time in 1944. It combines the advantages of both the steepest descent method (SDM) and the Gauss Newton (GNM) algorithm. Therefore, its convergence rate is higher than both SDM and GNM algorithms. This algorithm is robust, fast, and requires less memory as compared to other backpropagated algorithms available in the MATLAB neural networks toolbox. This backpropagated neural network algorithm is successfully implemented for nanofluidic systems, heat transfer effects, porous fin heat sink, magnetohydrodynamic systems, and pantograph delay systems [81,82,83,84,85].

### 2.3. Bayesian Regularization Backpropagation

Backpropagated neural networks with Bayesian Regularization are more robust than other optimization backpropagated neural networks. Therefore, in this paper, the Bayesian Regularization backpropagated algorithm is also used to find an optimized solution for the Dahl hysteresis model of the piezoelectric actuator. It is a numerical process that alters a nonlinear regression into a “well posed” statistical issue in the same way as ridge regression. In this optimization algorithm, the weight and bias values are updated until the desired output is achieved. It identifies the correct combination of squared errors and weights to construct a network that generalizes well by minimizing a combination of squared errors and weights. It also decreases or eliminates the requirements of lengthy cross-validation. In the proposed methodology, the implementation of this algorithm is carried out by using the neural networks toolbox available in the MATLAB platform for artificial intelligence paradigms. This backpropagated algorithm is successfully implemented for environmental economic systems [86].

### 2.4. Performance Indices

The performance analysis of suggested techniques for the Dahl-based hysteresis model has been determined by mean squared error, regression analysis, and curve fitting. The mathematical representation for performance matrices of the mean squared error and the correlation coefficient are shown in Equations (8) and (9).
(8)$$MSE=\frac{1}{s}\sum_{i=1}^{s}{\left(Y_{i}-{\overset{⌢}{Y}}_{i}\right)}^{2}$$
(9)$$R^{2}=1-\frac{\sum_{i=1}^{s}{\left({\hat{Y}}_{i}-{\bar{Y}}_{i}\right)}^{2}}{\sum_{i=1}^{s}{\left(Y_{i}-{\bar{Y}}_{i}\right)}^{2}}$$
where *s* represents the quantity of dataset points, and Y and $\overset{⌢}{Y}$ are real and predictable outputs, respectively. The error is acquired by comparing the predicted and real outcome values. *MSE* may be utilized as a cost function to examine the system’s functioning.

## 3. Results Interpretation

In this section, the numerical computational outcomes are presented for Dahl-based hysteresis nonlinear model PEAs using artificial neural networks. The two scenarios are defined based on the application of different backpropagated neural networks. In scenario 1, the characteristics of LMM-NNs are exploited, while in scenario 2, the potential of BRM-NNs is employed to obtain an approximate solution of the Dahl-based hysteresis model. Four cases of each scenario are defined based on distinct input voltage signals to actuate the piezoelectric model; a brief description is given below.

### 3.1. Case 1: Type 1 Input Signal for PEAs

In this case, the dynamic of Dahl-based hysteresis model characterized in (3) for PEAs is presented by considering the type 1 input voltage signal shown in Equation (10):(10)v(t)=sin(t),

Using Equation (10), the updated Dahl-based hysteresis model for the piezoelectric actuating system is specified in Equation (11).
(11)$$\left(m\ddot{d}(t)+\varepsilon \dot{d}(t)+gd(t)-k\sin(t)+a_{1}b_{1}-b_{0}a_{2}\mathrm{sgn}(\dot{d}(t))\right)=0.$$

### 3.2. Case 2: Type 2 Input Signal for PEAs

In case 2, the input voltage signal of type 2 presented in Equation (12) for time, $t\in \left[0,10\right]$, which is used to actuate the proposed system for the dynamics of the Dahl-based hysteresis model, is demonstrated in Equation (13).
(12)v(t)=50+50sin(πt),
(13)$$\left(m\ddot{d}(t)+\varepsilon \dot{d}(t)+gd(t)-50k-50k\sin(\pi t)+a_{1}b_{1}-b_{0}a_{2}\mathrm{sgn}(\dot{d}(t))\right)=0.$$

### 3.3. Case 3: Type 3 Input Signal for PEAs

In this case, a type 3 input voltage signal of time, $t\in \left[0,10\right]$, described in Equation (14), is elected for the piezoelectric actuating system to study the dynamics of the Dahl-based hysteresis model. The updated equation of the Dahl hysteresis model is demonstrated in Equation (15).
(14)$$v(t)=5e^{-0.13t}\left(\cos(3\pi te^{-0.09t}-3.15)+1\right)\;,$$
(15)$$\left(\begin{array}{l} m\ddot{d}(t)+\varepsilon \dot{d}(t)+gd(t)-b_{0}a_{2}\mathrm{sgn}(\dot{d}(t))-\\ 5e^{-0.13t}k\left(\cos(3\pi te^{-0.09t}-3.15)+1\right)+a_{1}b_{1} \end{array}\right)=0.$$

### 3.4. Case 4: Type 4 Input Signal for PEAs

In this case study, a Dahl-based hysteresis model shown in Equation (3) is demonstrated by selecting the type 4 input signal numerically shown in Equation (16). The updated Dahl-based hysteresis model is presented in Equation (17).
(16)$$v(t)=5+3.4e^{-0.24t}\left(\cos2\pi t\right)\;,$$
(17)$$\left(m\ddot{d}(t)+\varepsilon \dot{d}(t)-b_{0}a_{2}\mathrm{sgn}(\dot{d}(t))-5k-3.4e^{-0.24t}k\left(\cos2\pi t\right)+a_{1}b_{1}+gd(t)\right)=0.$$

Initially, the obtained datasets of all four cases for the Dahl-based hysteresis model of the piezoelectric actuating system are generated by the NDSolver function with the built-in Adams numerical technique in the Mathematica simulation environment. All four input voltage signals are graphically shown in Figure 4. The step size 0.001 is considered to create datasets for each case analysis to obtain the optimized solution of the dynamic Dahl-based hysteresis model backpropagated with LMM-NNs and BRM-NNs. The created dataset for each case contains 10,001 total data points. The sample datasets for four distinct cases having various voltage inputs for specific data points with a fixed step size of 0.5 are given in Table 2. Moreover, all these obtained datasets were transferred into the computational platform MATLAB for the execution of LMM-NNs and BRM-NNs, where 70% of the total data points were utilized for model training, 15% for model testing, and the remaining 15% were utilized for the validation process in order to achieve the desired results. The basic architecture of artificial neural networks contains three main layers: the input layer, hidden layer, and output layer, where the hidden layer neurons are kept at 100 to obtain efficient results.

The performance analysis based on mean squared error (*MSE*), gradient, step size values (Mu), epoch, and computational time for all four cases of the scenario 1 Dahl-based hysteresis PEA model is expressed in Table 3. The best performance value obtained in case 1 is 5.61^−6^, achieved in 9 s at the 194th epoch with step size 1^−8^ and gradient 5.32^−5^, for case 2 is 3.76^−6^ obtained in 12 s at the 100th epoch with step size 1^−8^ and gradient 1.49^−5^, for case 3 is 3.92^−7^ observed in 11 s at the 96th epoch with step size 1^−9^ and gradient 1.68^−4^, and for case 4 is 3.85^−6^ achieved in 14 s at the 99th epoch with step size 1^−9^ and gradient 1.76^−5^. Furthermore, the near-optimal mean squared error values achieved by model training, model testing, and model validation for case 1 are 5.65068^−6^, 1.15930^−5^, and 6.69292^−6^; for case 2 are 3.79119^−6^, 1.18214^−5^, and 1.47843^−5^; for case 3 are 9.32313^−7^, 1.21842^−6^, and 5.13605^−5^; and for case 4 are 3.88760^−6^, 1.46628^−5^, and 1.13105^−5^, respectively, which validate the system accuracy.

Performance analysis of all four cases for BRM-NNs of the proposed dynamic Dahl-based hysteresis model is demonstrated in Table 4. Case 1’s best performance value is 9.37^−8^ achieved in 151 s, that of case 2 is 5.53^−10^ obtained in 129 s, that of case 3 is 4.61^−6^ observed in 166 s, and that of case 4 is 9.64^−8^ achieved in 129 s. The best *MSE* values achieved for training and testing of case 1 are 9.36911^−8^ and 1.2342^−8^, for case 2 are 5.53105^−10^ and 8.6431^−11^, for case 3 are 4.61074^−6^ and 1.5354^−6^, and for case 4 are 9.63888^−8^ and 1.1736^−7^, respectively. The *MSE* values for validation are zero in BRM-NNS. Case 2 illustrates the optimal mean squared error value for system testing, which is 8.6431^−11^ with Mu of value 5, as compared to the other three cases.

The obtained results for both LMM-NNs and BRM-NNs are also presented in graphical form for all four cases. The neural network training states for validation fails, step size (mu), and the gradient are demonstrated graphically in Figure 5 and Figure 6. The fitting curves of output, target, and model error against each input are plotted in Figure 7 and Figure 8. Performance analyses are shown in Figure 9 and Figure 10. Regression analyses of output versus target values for model training, validation, and testing are presented in Figure 11 and Figure 12.

The state transitions that show the algorithm stability of all four cases for both LMM-NNs and BRM-NNs for the Dahl-based hysteresis model are graphically shown in Figure 6 and Figure 7, where the step size of algorithm Mu, number of parameters at each epoch, validation check of the model, and gradient are presented. The gradient values obtained for LMM-NNs are about [5.3^−5^, 1.4^−5^, 1.6^−4^, and 1.7^−5^] with step size Mu [1^−8^, 1^−8^, 1^−9^ and 1^−8^] and for BRM-NNs are [2.2^−6^, 1.7^−5^, 2.1^−5^ and 1^−6^] with step size Mu [5,50]. These results show the convergent and accurate performance of the proposed techniques LMM-NNS and BRM-NNs for all four cases of the Dahl-based hysteresis model PEAs.

The function fitness curve of all four cases for both LMM-NNs and BRM-NNs are illustrated graphically to obtain the accuracy of the Dahl-based hysteresis dynamic model of the piezoelectric actuating system, and hence the error has been found for all cases.

In function fitness curves for LMM-NNs, the output and target points are plotted for all training, testing, and validating concerning the input. For BRM-NNs, the output and target points are plotted for training and testing in order to approximate the solution shown in Figure 8 and Figure 9.

The performance analyses of four cases for both the LMM-NNs and BRM-NNs Dahl-based hysteresis models are graphically illustrated in Figure 10 and Figure 11. In these graphs, the mean squared error values are plotted across every epoch for all training, validation, and testing of a system model. The superlative validation performance obtained in case 3 of LMM-NNs is 1.2184 ^−6^ at the 90th epoch, while the best training performance is achieved in case 2 of BRM-NNs, which is 5.5311^−10^ at the 1000th epoch. The best line for all four cases of the LMM-NNs model is around 10^−5^, while the best lines for all four BRM-NN cases are 10^−8^, 10^−9^, 10^−5^, and 10^−7^. These graphs illustrate that the accurate *MSE* values were taken for training and validation; however, some degradation occurred in the testing of the Dahl-based hysteresis model due to unbiasedness. Moreover, at the input stage, the target was not defined during validation and testing.

The regression analyses of the Dahl-based hysteresis model for training, validation, testing, and for the total dataset was carried out using both LMM-NNs and BRM-NNs and are graphically shown in Figure 11 and Figure 12. The obtained results indicate a close correlation between target and output vectors by giving an *R* value approximately equal to one.

## 4. Conclusions

The potential of the multi-layered architecture of feedforward networks, i.e., LMM-NNs and BRM-NNs backpropagation neural networks, was applied to obtain precise, remarkable, efficient, and robust solutions of differential equations of a dynamic Dahl hysteresis model for the piezoelectric actuator. The Adams numerical method was used to develop the dataset for training, validation, and testing of the piezoelectric actuator model. Several variations were taken for conducting the simulation of the cases depending on the change in voltage signal used as an input to the piezoelectric actuator. The proposed methodology is implemented for all four cases of both scenarios based on LMM-NNs and BRM-NNs and we obtained the results with maximum accuracy with regard to a performance analysis by means of squared error and regression analyses. In scenario 1, case 1 illustrates the best *MSE* value for testing the proposed system, which is 6.7^−6^, achieved in the 194th epoch, while for BRM-NNs, case 2 illustrates the best *MSE* value for testing the presented system as compared to the rest of the cases, which is 8.6^−11^. Generally, the best performance was achieved by BRM-NN-based methodology.

## Figures and Tables

**Figure 1 micromachines-13-02205-f001:**
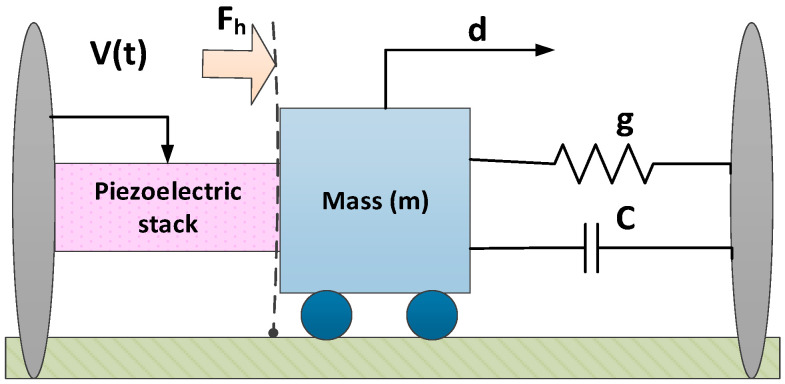
Pictorial representation of a piezoelectric actuator.

**Figure 2 micromachines-13-02205-f002:**
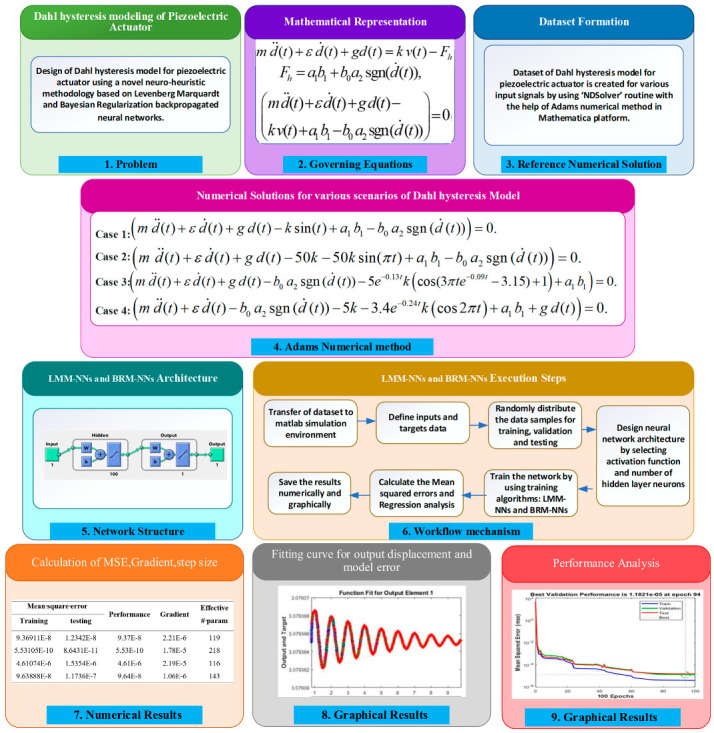
Graphical abstract for a proposed methodology of LMM-NNs and BRM-NNs for piezoelectric actuator.

**Figure 3 micromachines-13-02205-f003:**
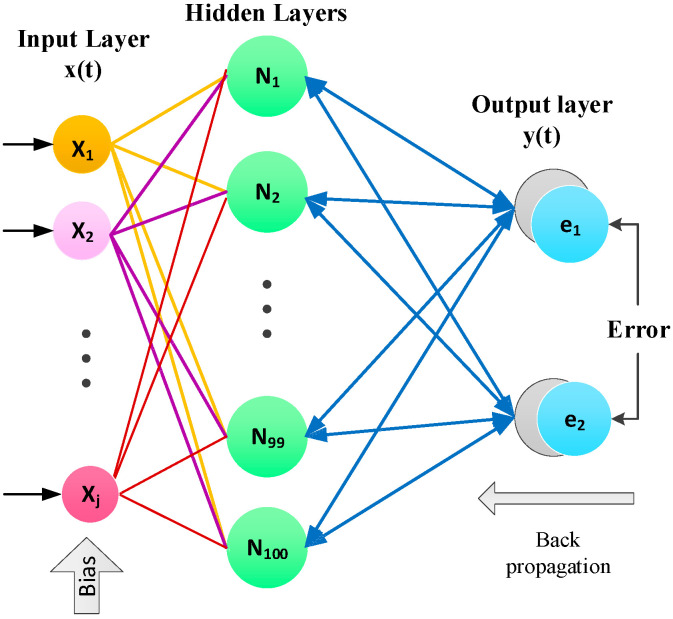
Structure of two layers of neural networks.

**Figure 4 micromachines-13-02205-f004:**
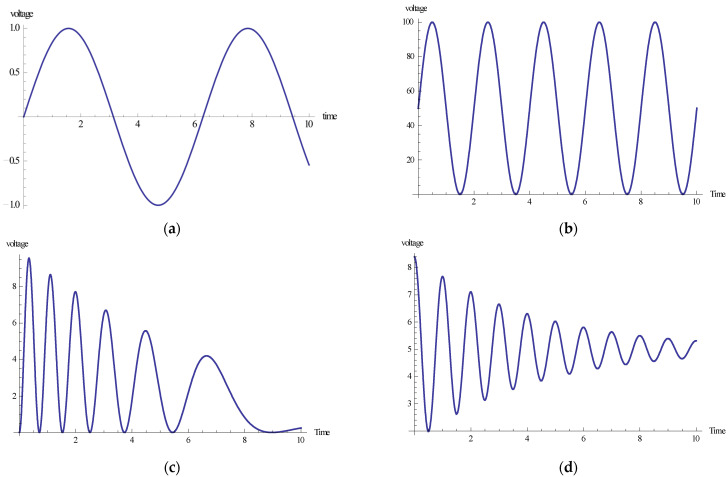
Input voltage signals of all four cases of both LMM-NNs and BRM-NNs. (**a**) Input Signal. (**b**) Input Signal 2. (**c**) Input Signal 3. (**d**) Input Signal 4.

**Figure 5 micromachines-13-02205-f005:**
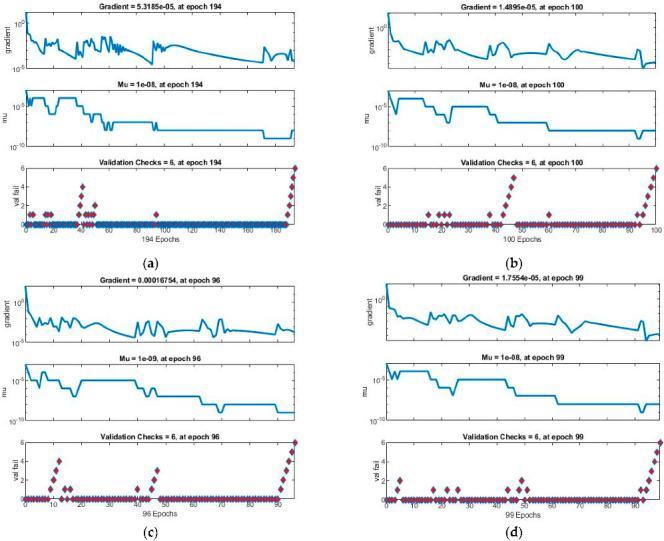
State transition containing gradient, step size, and validation checks of LMM-NNs for all four cases of the Dahl-based hysteresis model of piezoelectric actuator. (**a**) LMM-NNs Case 1. (**b**) LMM-NNs Case 2. (**c**) LMM-NNs Case 3. (**d**) LMM-NNs Case 4.

**Figure 6 micromachines-13-02205-f006:**
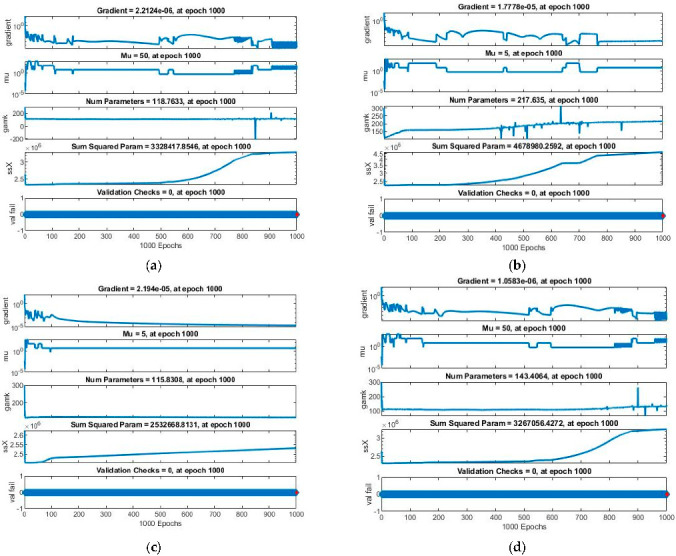
State transition containing gradient, step size, and validation checks of BRM-NNs for all four cases of the Dahl-based hysteresis model of piezoelectric actuator. (**a**) BRM-NNs Case 1. (**b**) BRM-NNs Case 2. (**c**) BRM-NNs Case 3. (**d**) BRM-NNs Case 4.

**Figure 7 micromachines-13-02205-f007:**
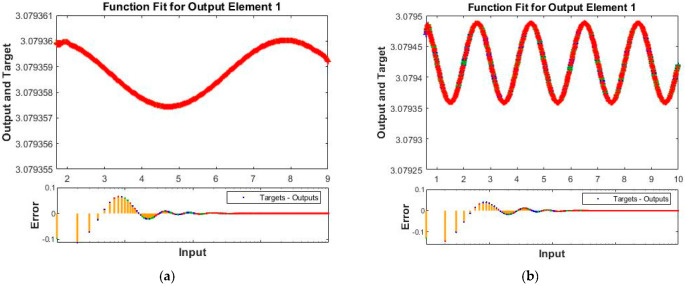
Function fitness curves for target, output elements, and error of all four cases of LMM-NNs for piezoelectric actuator. (**a**) LMM_NNs Case 1. (**b**) LMM_NNs Case 2. (**c**) LMM-NNs Case 3. (**d**) LMM-NNs Case 4.

**Figure 8 micromachines-13-02205-f008:**
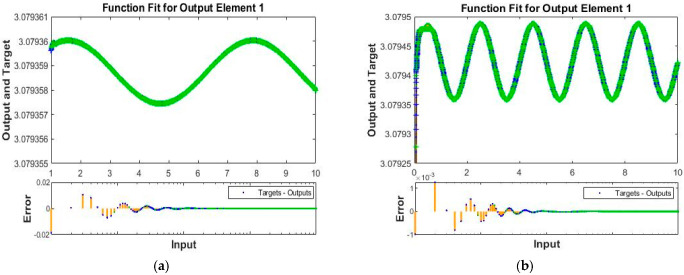
Function fitness curves for target, output elements, and error of all four cases of BRM-NNs piezoelectric actuator. (**a**) BRM-NNs Case 1. (**b**) BRM-NNs Case 2. (**c**) BRM-NNs Case 3. (**d**) BRM-NNs Case 4.

**Figure 9 micromachines-13-02205-f009:**
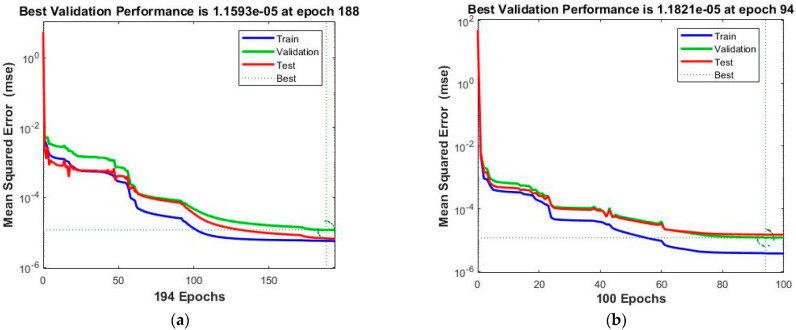
Performance analysis of train, validation, test, and best by LMM-NNs and BRM-NNs for all four cases of the Dahl-based hysteresis model. (**a**) LMM-NNs Case 1. (**b**) LMM-NNs Case 2. (**c**) LMM-NNs Case 3. (**d**) LMM-NNs Case 4.

**Figure 10 micromachines-13-02205-f010:**
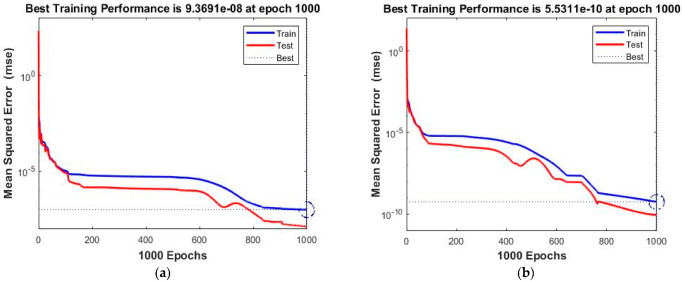
Performance analysis of train, validation, test, and best by BRM-NNs for all four cases of the Dahl-based hysteresis model. (**a**) BRM-NNs Case 1. (**b**) BRM-NNs Case 2. (**c**) BRM-NNs Case 3. (**d**) BRM-NNs Case 4.

**Figure 11 micromachines-13-02205-f011:**
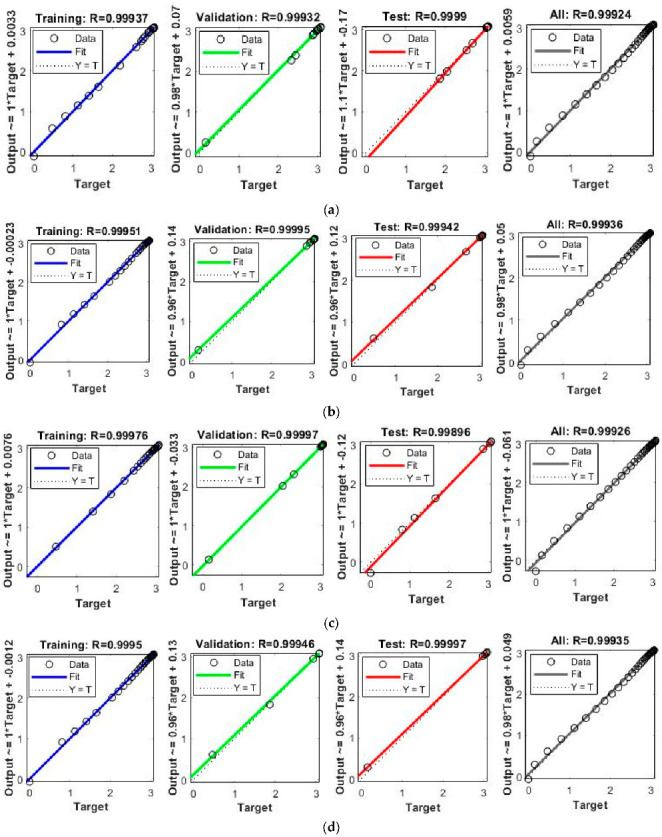
Regression analysis of training, validation, and testing for LMM-NNs of the Dahl-based hysteresis model of piezoelectric actuator. (**a**) LMM-NNs Case 1. (**b**) LMM-NNs Case 2. (**c**) LMM-NNs Case 3. (**d**) LMM-NNs Case 4.

**Figure 12 micromachines-13-02205-f012:**
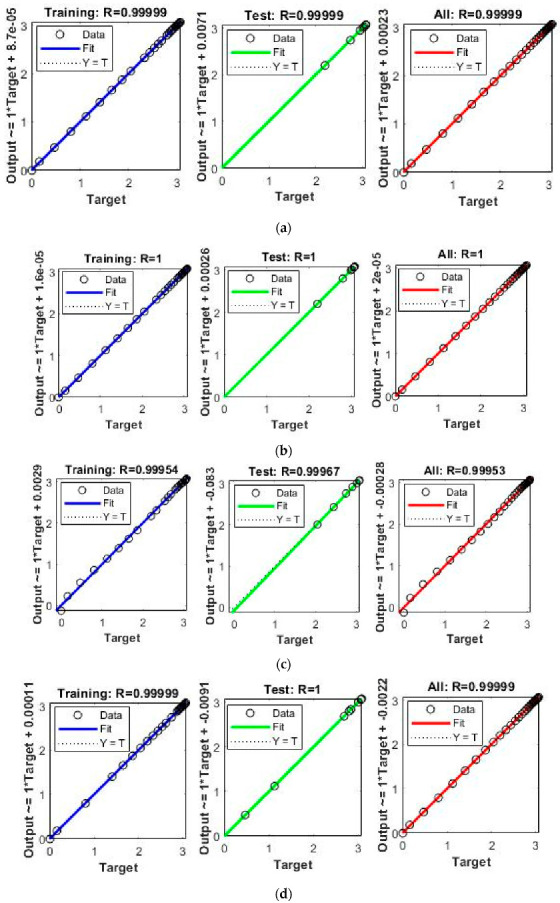
Regression analysis for training, validation, and testing by BRM-NNs of the Dahl-based hysteresis model of the piezoelectric actuator. (**a**) BRM-NNs Case 1. (**b**) BRM-NNs Case 2. (**c**) BRM-NNs Case 3. (**d**) BRM-NNs Case 4.

**Table 1 micromachines-13-02205-t001:** Values for parameters of the piezoelectric actuator.

Parameter	Value
m	0.1828 Kg
ε	190.154 Ns/m
*g*	2.6 × 10^4^ N/m
*k*	0.0336 C/m
b_1_	3.09292 × 10^5^
b_0_	0
a_1_	−0.25886
a_2_	7.0626

**Table 2 micromachines-13-02205-t002:** Output displacement (m) datasets of all cases of both scenarios.

t	Case-1	Case-2	Case-3	Case-4
	$d_{1}(t)$	$d_{2}(t)$	$d_{3}(t)$	$d_{4}(t)\;$
0.0	0.0000000	0.0000000	0.0000000	0.0000000
0.5	3.0793593	0.3080651	3.0793665	3.0793613
1.0	3.0793598	0.3080020	3.0793680	3.0793687
1.5	3.0793600	0.3079359	3.0793589	3.0793621
2.0	3.0793599	0.3079990	3.0793687	3.0793679
2.5	3.0793595	0.3080651	3.0793588	3.0793628
3.0	3.0793589	0.3080020	3.0793670	3.0793673
3.5	3.0793582	0.3079359	3.0793610	3.0793633
4.0	3.0793577	0.3079990	3.0793609	3.0793669
4.5	3.0793574	0.3080651	3.0793659	3.0793637
5.0	3.0793574	0.3080020	3.0793615	3.0793665
5.5	3.0793578	0.3079359	3.0793588	3.0793640
6.0	3.0793583	0.3079990	3.0793615	3.0793662
6.5	3.0793590	0.3080651	3.0793640	3.0793643
7.0	3.0793595	0.3080020	3.0793635	3.0793660
7.5	3.0793599	0.3079359	3.0793616	3.0793645
8.0	3.0793600	0.3079990	3.0793598	3.0793658
8.5	3.0793597	0.3080651	3.0793589	3.0793646
9.0	3.0793592	0.3080020	3.0793587	3.0793657
9.5	3.0793586	0.3079359	3.0793589	3.0793647
10	3.0793580	0.3079990	3.0793591	3.0793656

**Table 3 micromachines-13-02205-t003:** Performance analysis of LMM-NNs for the proposed model.

Case	Mean Squared Error			-Performance	-Gradient	-Mu	Epoch	Time (s)
	-Training	-Validation	-Testing					
1	5.65068^−6^	1.15930^−5^	6.69292^−6^	5.61^−6^	5.32^−5^	1^−8^	194	09
2	3.79119^−6^	1.18214^−5^	1.47843^−5^	3.76^−6^	1.49^−5^	1^−8^	100	12
3	9.32313^−7^	1.21842^−6^	5.13605^−5^	3.92^−7^	1.68^−4^	1^−9^	96	11
4	3.88760^−6^	1.46628^−5^	1.13105^−5^	3.85^−6^	1.76^−5^	1^−8^	99	14

**Table 4 micromachines-13-02205-t004:** Performance analysis of BRM-NNs for the proposed model.

Case	Mean Squared Error		Performance	Gradient	Effective Param	Sum Squared Param	Mu	Epoch	Time (s)
	-Training	-Testing							
1	9.36911^−8^	1.2342^−8^	9.37^−8^	2.21^−6^	119	3.33^6^	50	1000	151
2	5.5311^−10^	8.6431^−11^	5.53^−10^	1.78^−5^	218	4.68^6^	5	1000	129
3	4.61074^−6^	1.5354^−6^	4.61^−6^	2.19^−5^	116	2.53^6^	5	1000	166
4	9.63888^−8^	1.1736^−7^	9.64^−8^	1.06^−6^	143	3.27^6^	50	1000	129

## Data Availability

Not applicable.

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
