# Peer review of "Intelligent Predictive Solution Dynamics for Dahl Hysteresis Model of Piezoelectric Actuator"

_micromachines, 2022, doi:10.3390/mi13122205_

Round 1

Reviewer 1 Report

In general, presentation of the manuscript and descriptions are made well, however some claims not cogent and descriptions in pages 10 and 11 are difficult to follow.

In section 4 (conclusion), the authors claimed that several experiments are undertaken.     I do not see any experimental setup in the manuscript to support this claim. In page 9 and in the conclusion, we understand that the data sets are generated numerically. I think the manuscript should be revised accordingly.

 Some of the refences cited in the manuscript should be citing the original studies. When talking about Dahl Model, one would expect to see the original articles, but not the papers refereeing (or citing) to Dahl model. Below reference can be an alternative to reference [21] or can be cited together. (DAHL J. R ., 1976. Solid friction damping of mechanical vibrations. A.I.A.A. J .. 14.n• 12. dec .. 1675-1682.)

 I think readability would be improved if the standard symbols are used for some parameters such as for piezoelectric coefficient and mass etc.

 In the text, many words are divided with a minus (-) sign (probably due to a format change) and sometimes punctuation marks are missing. All these readability disturbances should be revised.

 Figure captions can be more descriptive (such as axis labels and units) especially for figures 6-13.

To see the comparisons clearly, maximum, and minimum values of the figures should be set to same values. This is important especially for error plots.     

 Either reference [85] is missing or cited reference number is wrong (page 7) and ref [61] is not cited correctly in page 3.

I think, the first sentence of the abstract should begin as; “Piezoelectric actuated systems…” rather than “Piezoelectric actuated models…”.

Y_bar (probably mean values) is not described after equation (9).

Can the authors comment about the system response for different input signals?

In practical application, how the proposed modeling can be used for different inputs?

Would there be a frequency limitation due to calculation time?

Author Response

Response to Reviewer 1 Comments

Response to the comments on the submitted manuscript ID: micromachines-2061455

Paper Title: Intelligent Predictive Solution

Dynamics for Dahl Hysteresis Model of Piezoelectric Actuator

Reviewer 1 General Comment:

In general, presentation of the manuscript and descriptions are made well, however some claims not cogent and descriptions in pages 10 and 11 are difficult to follow.

Response: The authors would like to thank anonymous reviewer for their valuable, detailed and constructive comments that allow us to further improve the quality of our submitted manuscript. All the comments and suggestions are carefully addressed in this document and the paper is accordingly revised.

Point 1: In section 4 (conclusion), the authors claimed that several experiments are undertaken. do not see any experimental setup in the manuscript to support this claim. In page 9 and in the conclusion, we understand that the data sets are generated numerically. I think the manuscript should be revised accordingly.

Response: Agreed with the reviewer that no particular hardware based experiement is conducted for the presented study, while the presented study is based on software based numerical study of dahl hysteresis model of piezoelectric actuator by using artificial neural networks with trainig algorithms like, Bayesian regularization and Levenberg Marquardt backpropagation algorithms. Therefore, we have updated the manuscript by modifying the conclusion section in order to rectify the ambiguity if any for better understanding fo the readers as sugggested.

Point 2: Some of the references cited in the manuscript should be citing the original studies. When talking about Dahl Model, one would expect to see the original articles, but not the papers refereeing (or citing) to Dahl model. Below reference can be an alternative to reference [21] or can be cited together. (DAHL J. R ., 1976. Solid friction damping of mechanical vibrations. A.I.A.A. J .. 14.n• 12. dec .. 1675-1682.)

Response: Agreed, We have updated the reference section of the revised manuscript by citations of original studies in the body of the draft for better understanding as suggested. Additionally, the mentioned/pointed relevant citations have now been added in the revised manuscript. Futhermore, authors has double checked each citation mentioned in the body of the manuscirpt and only those reference are given which are relevant to the study conducted in this research.

Point 3: I think readability would be improved if the standard symbols are used for some parameters such as for piezoelectric coefficient and mass etc.

Response: Agreed with the statement “I think readability would be improved if the standard symbols are used for some parameters” of the worthy anonymous reviewer. Thanks for the valuable suggestion. Now the standard symbols of piezoelectric parameters and mass have been used in the revised manuscript as suggested for better understanding and readability.

Point 4: In the text, many words are divided with a minus (-) sign (probably due to a format change) and sometimes punctuation marks are missing. All these readability disturbances should be revised.

Response: Agreed. We have updated the whole manuscirpt after careful reading to rectify the issue of (-) comes after format change and also corrected the punctuation marks as suggested for better understanding of the readers.

Point 5: Figure captions can be more descriptive (such as axis labels and units) especially for figures 6-13.

Response: Agreed, We have updated the manuscirpt after critical, careful and detailed review of all Figure captions and now provided with more-descriptive manner as suggested by the worthy anonymousr reviewer for better understanding.

Point 6: To see the comparisons clearly, maximum, and minimum values of the figures should be set to same values. This is important especially for error plots.

Response: Agreed, We have updated the manuscirpt after careful review of graphical illustrations for the error plots as suggested. The values of these errors, i.e., MSE for training testing, and validation are provided in Table 3 for all four cases of the system model via AI based computing procedure of LMM-NNs , while these results of BRM-NNs are provided in Table 4 for each variation of the system model.

Point 7: Either reference [85] is missing or cited reference number is wrong (page 7) and ref [61] is not cited correctly in page 3.

Response: Agreed. We have updated the mansucirpt after careful review of all citation mentioned in the body of the draft as suggested. Moreover, we have rectified the pointed concerns in the references [85] (now [86]) and [61] (now [63]) in the revised manuscirpt. Thanks for pointing out this ambiguity in the text, All the mentioned references have now been cited corectly in the revised manuscript.

Point 8: I think; the first sentence of the abstract should begin as; “Piezoelectric actuated systems…” rather than “Piezoelectric actuated models…”.

Response: Thanks for the valuable suggestion. The first sentence of abstract has now been updated according to the suggestions of worthy anonymous reviewer for better understanding and readability.

Point 9: Y_bar (probably mean values) is not described after equation (9).

Response : Agreed, is the predictable output in equation (9), which has now been added in the revised manuscript.

Point 10: Can the authors comment about the system response for different input signals?

Response: Different input voltage signals were applied in order to check the generalization of the presented models LMM-NNs and BRM-NNs based networks for dahl hysteresis model of piezoelectric actuators. The considering of the input signal variations allows to capture the rate-dependency of the hysteresis models.

Point 11: In practical application, how the proposed modeling can be used for different inputs?

Response: The hysteresis of piezoelectric actuators depends upon the amplitude as well as the frequency of input signals. Hence the varaition of input signal allows to capture the rate dependency of the hysterical effect. They are used in applications: where fast response (high frequency) is required with small displacement amplitude, and slow response (low frequency) with smooth motion is required. Practically they are used in high precision applications like micro/nano positioning systems, scanning probe microscopes, lithography, atomic force microscopic probes etc

Point 12: Would there be a frequency limitation due to calculation time?

Response: The hysterical effect of piezoelectric actuators depends upon the frequency of the input signal. Hence variation of input sine wave signal frequency cause variation in output hysteresis. The input signal with higher frequency will have greater hysterical effect.

Reviewer 2 Report

This paper proposes an artificial intelligence-based neurocomputing feedforward and backpropagation networks of the Levenberg–Marquardt method and Bayesian Regularization method to examine the numerical behavior of the Dahl hysteresis model representing piezoelectric actuator. Although the paper is interesting, the following issues should be addressed.

1) It may be more mainstream to abbreviate Piezoelectric Actuators to PEAs.

2) The review of hysteresis models is incomplete in the introduction. Since this paper focuses on the artificial intelligence-based model, some data-driven models based on neural networks should also be discussed, e.g., the gated recurrent unit based frequency-dependent hysteresis model, the improved neural Turing machine based frequency-dependent hysteresis model, etc.

3) It is inappropriate to directly use the MATLAB screenshot in Figure 3. It may be better to redraw the structure of two layers of neural network.

4) There are some grammatical errors in the paper, the paper needs careful proof-reading.

Author Response

Response to Reviewer 2 Comments

Response to the comments on the submitted manuscript ID: micromachines-2061455

Paper Title: Intelligent Predictive Solution

Dynamics for Dahl Hysteresis Model of Piezoelectric Actuator

Reviewer 2 General Comment:

This paper proposes an artificial intelligence-based neurocomputing feedforward and backpropagation networks of the Levenberg–Marquardt method and Bayesian Regularization method to examine the numerical behavior of the Dahl hysteresis model representing piezoelectric actuator. Although the paper is interesting, the following issues should be addressed.

Response: The authors would like to thank anonymous reviewer for his/her valuable, detailed and constructive comments that allow us to further improve the quality of our submitted manuscript. All the comments and suggestions are carefully addressed in this document and the paper is accordingly revised.

Point 1: It may be more mainstream to abbreviate Piezoelectric Actuators to PEAs.

Response: Thank you so much for this suggestion, the Piezoelectric Actuators has now been abbreviated as PEAs in the body of the revised manuscript for better understanding and readability.

Point 2: The review of hysteresis models is incomplete in the introduction. Since this paper focuses on the artificial intelligence-based model, some data-driven models based on neural networks should also be discussed, e.g., the gated recurrent unit based frequency-dependent hysteresis model, the improved neural Turing machine based frequency-dependent hysteresis model, etc.

Response: Agreed. We have updated the manuscirpt by conducting the detailed review of the introduction section in order to provided the literature review and problem statement with latest state of the art journal paper as suggested for better understanding. Moreover, authors are thankful for providing the recent, relevant and reputed literature [31-32] on the topic of the journal. We have read these articles and definitely help a lot to improve the technical and presentation strength of the introudcitoin section of the revised manuscript.

Please see the updated introduction and reference sections  of the revised manuscript.

Point 3: It is inappropriate to directly use the MATLAB screenshot in Figure 3. It may be better to redraw the structure of two layers of neural network.

Response: Agreed, We have removed the pointed figures from the body of the manuscrip Additionally, we have provided the figure 3 based on two layers structure of the neural networks in the updated manuscript as follows:

Figure 3. Structure of two layers of neural networks

Point : There are some grammatical errors in the paper, the paper needs careful proof-reading.

Response: Agreed. We have updated the manuscript by careful, critical and exhaustive reviewe of the whole draft to improve the liguistic quality by avoiding the grammatical/topographical error, ambiguous sentences and punctuations mistakes for better understanding of the readers as suggested by the worthy anonymous reviewer. Moreover, the English/Language corrections, we have requested the most respected Professor Dr. Dumitru Baleanu, Cankaya University, Turkey, and Institute of Space Sciences, Bucharest, Romania. We appreciate his help, efforts and support for the improvement of the revised manuscript.